

# Methodology for the assessment of poor-data water resources

María del Mar Navarro-Farfán[1], Liliana García-Romero[2],
Marco A. Martínez-Cinco[1], Constantino Domínguez-Sánchez[2] and
Sonia Tatiana Sánchez-Quispe[2]

[1] Faculty of Chemical Engineering, Universidad Michoacana de San Nicolás de Hidalgo, Morelia, Michoacán, Mexico
[2] Faculty of Civil Engineering, Universidad Michoacana de San Nicolás de Hidalgo, Morelia, Michoacán, Mexico

Corresponding authors
Liliana García-Romero,
liliana.romero@umich.mx
Sonia Tatiana Sánchez-Quispe,
quispe@umich.mx

## ABSTRACT

Surface hydrologic modeling becomes a problem when insufficient spatial and temporal information is available. It is common to have useful modeling periods of less than 15 years. The purpose of this work is to develop a methodology that allows the selection of meteorological and hydrometric stations that are suitable for modeling when information is scarce in the area. Based on the scarcity of data, a series of statistical tests are proposed to eliminate stations according to a decision-making process. Although the number of stations decreases drastically, the information used is reliable and of adequate quality, ensuring less uncertainty in the surface simulation models. Individual basin modeling can be carried out considering the poor data. The transfer of parameters can be applied through the nesting of basins to have information distributed over an extensive area. Therefore, temporally and spatially extended modeling can be achieved with information that preserves statistical parameters over time. If data management and validation is performed, the modeled watersheds are well represented; if this is not done, only 26% to 50% of the runoff is represented.

# INTRODUCTION

Hydrographic basins respond to climatic, geographic, and anthropogenic changes as a result of the spatial and temporal variation of climate and environmental conditions. Lack of long-term data, differences between databases, and data collection complicate the spatial and time series analysis needed for modeling (*Pilgrim et al., 2015*).

Hydrologic models of surface water require characteristics of land use and soil type, topography (*Zhao, Zhang & Cheng, 2018*), and climatic data (*Ang & Oeurng, 2018*; *Wright, 2018*). Therefore, high-quality climate information is essential for successful and reliable modeling (*Wright, 2018*).

However, the span between rainfall observation and its availability in a database implies a lack of extended and consistent data recording. In regions with scarce data, the modeling process, calibration, and subsequent validation are challenging due to the lack of reliable
information (*Aduah, Jewitt & Warburton Toucher, 2017*). Regardless of the model's simplicity, the input data is fundamental to its accuracy.

Modeling hydrology in regions with limited and uncertain information is a reality (*Arellano-Lara & Escalante-Sandoval, 2014*). It should be considered that the desirable period for robust modeling is 30 years (*WMO, 1989*); however, the databases worldwide and in Latin America have deficient data with short information series, which is why the modeling must be done with scarce information.

Methods and models employed are contingent upon available information, and in some cases, deficiencies in observation data are compensated for through robust modeling techniques (*Quintana-Seguí et al., 2020*). Notably, data series spanning up to 13 years, often without updates, have been utilized by *Hammer & Kadlec (1986)*.

The limited availability of data has led to the implementation of models with extremely short periods, such as 4 and 2 years for calibration and validation, respectively (*Ang & Oeurng, 2018*). Although data scarcity is a common issue, there is currently no established methodology to help improve the reliability of poor-data simulations.

However, there are precedents of successful modeling with shorter periods ranging from 6 to 15 years (*Marcinkowski & Mirosław-Świątek, 2020*; *Wielgat et al., 2021*; *Goshime et al., 2021*; *Mehla, 2022*). Moreover, literature suggests modeling periods varying from 15 to 20 years (*Khoshkhoo et al., 2015*; *Rasoulzadeh Gharibdousti, Kharel & Stoecker, 2019*; *Dehghanipour et al., 2019*; *Lerat et al., 2020*; *Bazzi, Ebrahimi & Aminnejad, 2021*; *Nazeer et al., 2022*), highlighting the preference for extended modeling periods (*Schuol et al., 2008*; *Li et al., 2015*; *Adla, Tripathi & Disse, 2019*; *García-Romero et al., 2019*; *Lakshmi & Sudheer, 2021*) that include validation and warm-up spans as well as modeling for complete or discontinuous periods (*Zhang et al., 2011*).

According to *Dehghanipour et al. (2019)*, proper modeling should consider a warm-up, calibration, and subsequent validation period (*García-Romero et al., 2019*), with at least 10 years for each stage.

The former works do not mention the methodology for data management and processing. Thus, there is no established method concerning the tests performed in the information validation stage and the selection of the time series from the meteorological stations (precipitation measurement) or hydrometric stations (flow records) to be used for modeling.

The motivation for this research arises from the need to manage data's temporal and spatial scarcity in order to perform hydrological modeling representative of the area. It is important to put considerable effort into collating and validating a regional hydro-meteorological database (*Johnston & Smakhtin, 2014*). Not only is the lack of information of concern, but it is also relevant to consider that it is inadequate to have isolated data in the study area.

Currently, the scarcity of information is a real problem that must be approached from a data management point of view so that modeling can be extrapolated to recent years or even to future periods. Therefore, it is proposed to carry out a complete prior review of the information to improve modeling reliability despite short periods.

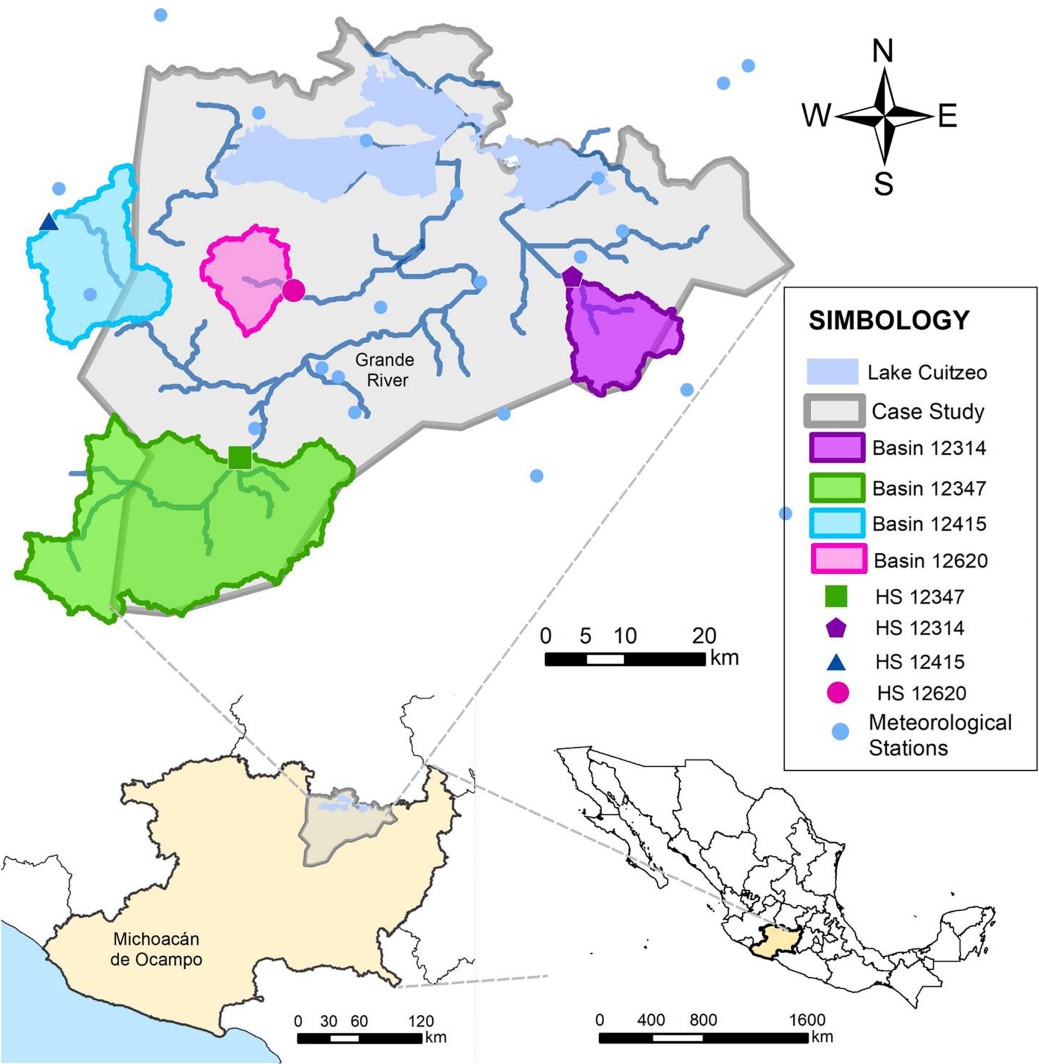

**Figure 1  The geographical location of the study area, discretization of the modeled basins, and location of the hydrometric and meteorological stations.** HS, hydrometric station. The authors' elaboration by means of QGis3.18.

Addressing information scarcity requires a comprehensive review of data to enhance modeling reliability despite limited periods. This study employs a hydrological surface model utilizing carefully selected scarce data, adhering to a methodology tailored for deficient temporal and spatial information. Establishing methodologies for precipitation and flow series analysis is crucial for reducing uncertainty associated with modeling.

This article delineates the case study, methodology, results (in the study area in general and with greater detail in the most representative basin), and conclusions.

## Case study

The study area is the hydrological basin that corresponds to the Morelia-Queréndaro aquifer, which is located in central Mexico. It has an average annual precipitation of 841 mm. The maximum annual precipitation is 1,336 mm in the southern portion and minimum annual precipitation of 654 mm in the north of the basin. According to the

Köppen-Geiger classification, the climate is type Cwb, corresponding to a moderate climate with dry winters and warm summers. Lake Cuitzeo, one of the most important lakes in Mexico, is located to the north of the basin. This is an area with a significant decrease in water resources, and the studies that have been carried out reflect the scarcity of information about drought beginning after 1999 (*Mendoza et al., 2006*; *Williams, 2014*).

For evaluating the water resources of the entire basin four calibration points are defined according to the hydrometric stations shown in Fig. 1 (HS 12347, HS 12314, HS12415 and HS 12620). Only the results of hydrometric station 12415 are presented.

## MATERIALS AND METHODS

A general methodology is described in Fig. 2 to evaluate water resources with data scarcity. It is divided into two stages: the first is the review of the hydrological series to maintain or discard the stations that will be able to generate the basins that are going to be used as i) head basins and, ii) a comparative of the surface modeling. The second stage is referred to the meteorological data, this information is reviewed and extended using the IDW method (*Chen & Liu, 2012*) to obtain complete series for the input data in the surface modeling.

### Materials

The Soil Moisture Method model, available in the WEAP software (*Sieber & Raskin, 2001*), was used to develop this work. WEAP is a hydrological software with a semi-distributed application. It requires a homogeneous set of climatic data (precipitation, temperature, and geographic latitude) for each sub-basin, divided into different land cover and land-use types (*Sieber & Purkey, 2015*).

The edaphological and land use information in vector files was acquired from the database of the National Commission for the Knowledge and Use of Biodiversity (CONABIO) (*CONABIO, 2023*), which was then modified in a geographic information system (GIS). Meteorological and hydrometric information was downloaded from the online databases of the Climate Computing project (CLICOM) (*Sistema Meteorológico Nacional (SMN), 2023*) and the Banco Nacional de Datos de Aguas Superficiales (BANDAS) (*CONAGUA, 2021*), respectively.

### Analysis and validation of the data

The most important aspect of the methodology is the analysis and validation of the meteorological and hydrometric stations, which undergoes a selection process to determine their applicability.

### Meteorological stations

The validation of the meteorological information was carried out following the process shown in Fig. 3. It is important to emphasize that the purpose of the meteorological data is to obtain complete series that has to be robust for their use in surface modeling. Being robust, it meets the minimum requirements necessary to perform hydrological modeling that can be reproduced at different scales, either as individual or grouped basin modeling.

A radius of influence was defined for the spatial selection of the meteorological stations, considering up to an additional 60% of the maximum radius of the basin so that the

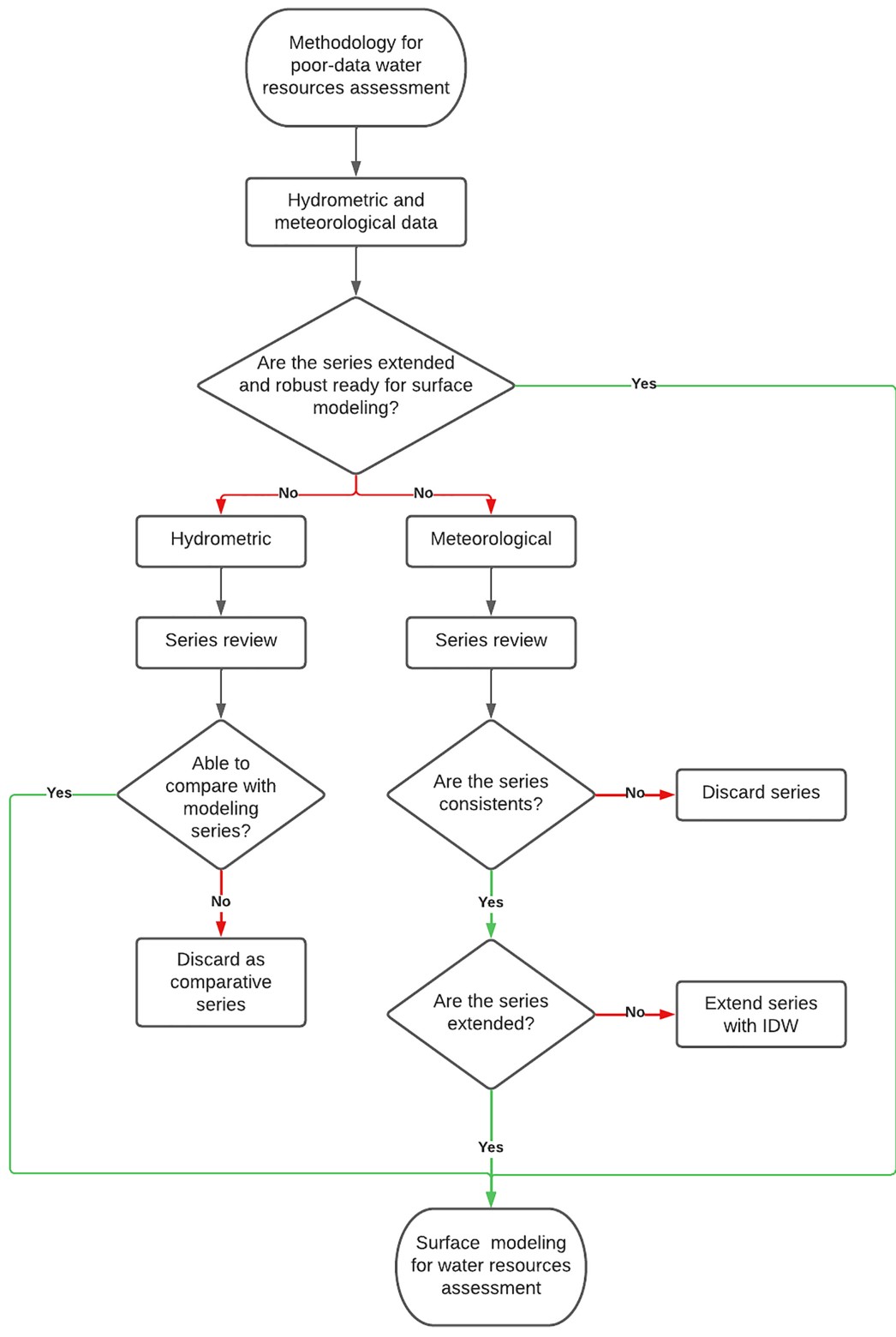

**Figure 2** **General methodological diagram for water resources assessment in data-scarce basins.**

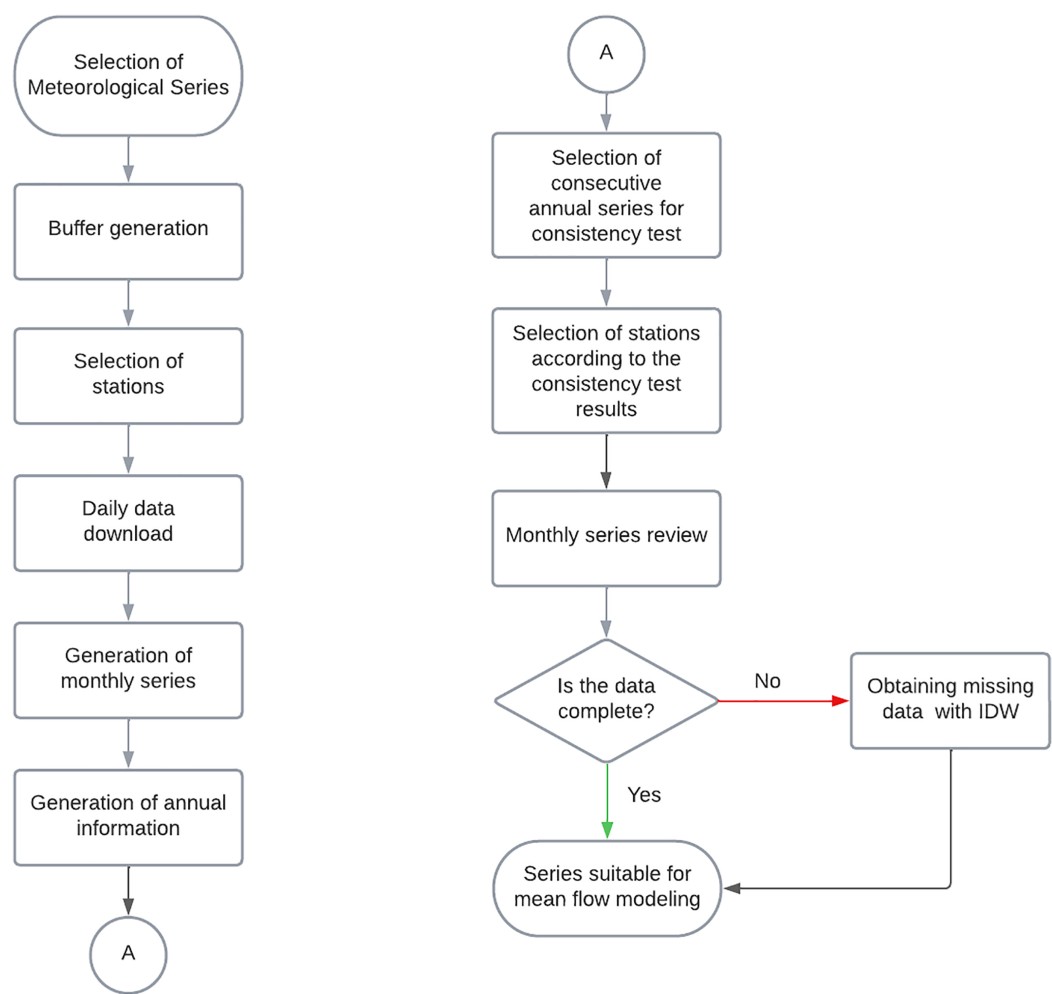

**Figure 3 Processing of information from meteorological stations.**

stations would cover the study area. This condition is validated with the Thiessen polygons (*Herschy & Fairbridge, 1998*).

The stations suitable for modeling that are within the influence radius are selected; for this purpose, the following criteria are established. First, the station is verified to be active and to have at least 30 years of information (*WMO, 1989*). Then, the percentage of gaps is obtained (a desirable value is <20% since these series should be completed in case of missing data, and having a high rate maintains a higher uncertainty). The studied period is also influential since the meteorological information must match the hydrometric data. Finally, the distance between the station and the center of gravity of the basin is considered using the Euclidean distance (*Arellano-Lara & Escalante-Sandoval, 2014*).

The generation of the time series is based on the analysis and validation of the daily scale data. Monthly scale series were generated from the daily data. It is necessary to have at least 21 days of information in the month, which are accumulated to obtain the monthly value,

to go from daily to monthly data. If there are not at least 21 days of information in each month, it is considered insufficient, and therefore the month will be null (empty).

The annual series is generated with the monthly series, and the first step is to obtain the monthly and annual averages. According to the annual precipitation contribution represented by the months, only up to two missing wet and dry months are allowed to consider the annual precipitation value as the sum of the precipitation of all months. The wettest month cannot be missing because it is the most representative month and if it is missing, the precipitation data for the whole year must be discarded. Annual series is developed and used for consistency tests.

Once the series is generated, tests are performed to evaluate homogeneity and independence, two essential characteristics of precipitation series that represent that the series is consistent. This is achieved through the implementation of general tests such as Sequences (*Mather, 1975*), the Helmert test (*Doorenbos, 1976*), the double Mass Curve (*Martínez, Martínez & Castaño, 2006*), and Wald-Wolfowitz (*Siegel, 2015*), in addition to specific tests such as Student's t-test (*WMO, 1966*), and Cramer (*Salas et al., 1980*). These tests demonstrated that the elements present in the sample come statistically from the same population.

Similarly, the rainfall series must demonstrate independence. This property is evaluated through the Anderson limits test (*Salas et al., 1980*). It is stated that a series is independent when the probability of occurrence of any precipitation data present in the sample does not depend on the occurrence of the subsequent or preceding precipitation value in space or time.

Regarding the results of the homogeneity tests, each station is expected to comply with the general tests. However, if any of these are unsuitable, the particular tests must be reviewed to ensure that the series is indeed homogeneous. On the other hand, the independence test must be complied with at all stations.

Precipitation series have gaps that must be filled using some data-filling method; for surface modeling, it is necessary to have series on a monthly scale. In this particular case, the IDW (inverse distance weighted) method is applied according to the Euclidean distance (*Chen & Liu, 2012*; *Arellano-Lara & Escalante-Sandoval, 2014*).

## Hydrometric stations

The selection and validation of the hydrometric stations must also be carried out. This procedure is shown in Fig. 4.

In the first stage, the information is downloaded, and the series is validated. Subsequently, the trend of the series is reviewed graphically. In the third stage, the consistency of the series is analyzed. And finally, characteristic parameters of the basin are obtained.

Once the study area is selected, the stations within the buffer must be chosen. Therefore, a buffer of 40% to 50% of the maximum radius of the basin is proposed. The buffer is limited because the hydrometric stations are required to discharge within the study area or be a part of the area.
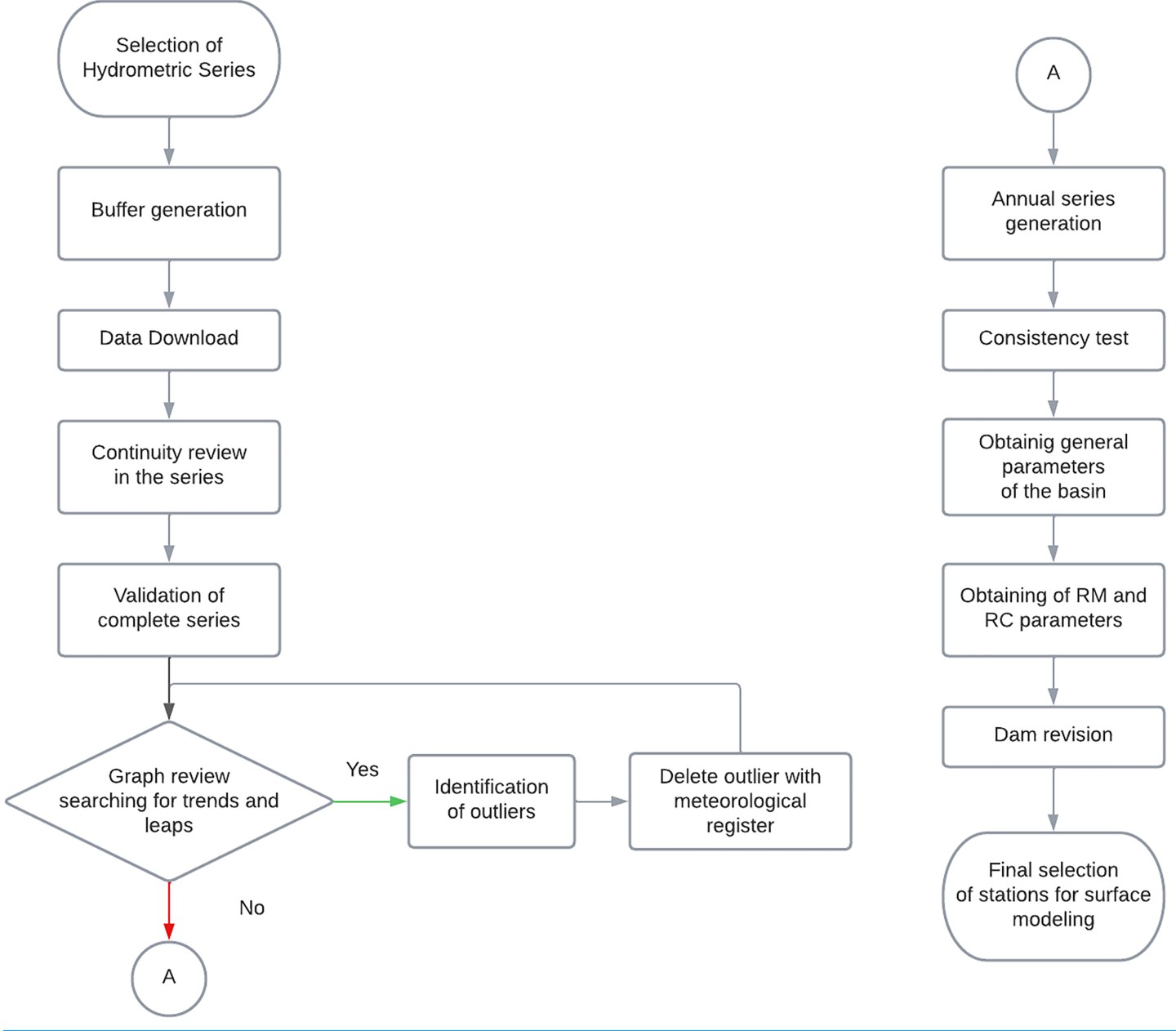

**Figure 4 Methodological scheme for hydrometric stations.**

The hydrometric information is downloaded every month, taking into account the continuity of the series. If there is at least 75% of the monthly data in a year, it is added to generate the annual series.

It is important to mention that for the hydrometric series, the estimation of missing data was not performed. It is proposed to use the longest series to obtain the information from the selected stations. In addition, it is suggested to have as much information as possible for the modeling, which reduces the uncertainty of the modeling expected to have a continuous period of at least 15 years. The estimation of missing data at hydrometric

stations cannot be done with a simple method. Therefore, the percentage of gaps must be obtained, and available information must be known.

Once the series is ordered, the longest complete and consecutive monthly series is selected to generate an annual series, with which the annual graphs will be made. These should be reviewed graphically to detect the presence of leaps and if there is an upward or downward trend, which can represent an alteration in the basin.

The presence of an alteration in the series is a valuable indicator of whether the basin is truly in a natural regime. This step is part of a review but is not a limiting factor.

Within the same graphic review, it is essential to check for the presence of atypical data. In case of any, it should be compared with the meteorological values to verify its accuracy. It is proposed to eliminate the outliers and redo the graphs to corroborate the trend or presence of leaps if the runoff value does not coincide with the precipitation. As a second graphical revision, the annual graphs should be created, accompanied by mobile means to observe their preservation.

Once the annual series is generated, consistency tests are performed, considering homogeneity and persistence. It is advised to use general homogeneity and persistence tests (Anderson limits). Homogeneity is also used to determine the trend since it is an excellent indicator of mean preservation. If it is not fulfilled, it can indicate that the series is inadequate. On the other hand, persistence in the series is desirable yet not a limiting characteristic. Due to the scarcity of hydrometric information, it is not possible to perform statistical tests such as the double mass curve or the Pettitt test.

The next step is to obtain the relative modulus (RM) (*Ortega et al., 1988*) and the runoff coefficient (RC) (*Campos-Aranda, 2014*), both indicators of the alteration that may occur in the basin. For the RM, it is proposed to use those stations that give values between three and 10 and, for the runoff coefficient, values less than one.

Compliance with the values established above for each parameter (RM and RC) indicates that the basin is in a natural regime and can therefore be used; otherwise, they should be discarded.

With the preceding recommendations, the number of stations (hydrometric and meteorological) to be used has been reduced. This selection ensures adequate stations and, therefore, better quality modeling.

## Modeling and model calibration

With the precipitation and flow data having been analyzed and validated, modeling is performed using the soil moisture method (SMM) in the WEAP software. The modeling is executed with the basins generated from the selected hydrometric stations, and since they do not cover the entire study area, the information is extrapolated.

Calibration is accomplished by varying four of the nine parameters involved in the SMM. The variation of the parameters is related to the application site. Within each model, a sensitivity analysis must be completed to determine the influence of each parameter on the model output. The calibrated parameters are: i) crop coefficient (Kc), which conditions the evapotranspiration volume and depends on the land use; ii) runoff resistance factor
(RRF) establishes the ratio of water that can runoff concerning the natural limiting factor of the soil; iii) soil water capacity (SWC) considers the moisture-holding capacity in the subsurface portion of the soil and; iv) preferred flow direction (F) marks the preferential direction of flow as if limited by the angle of infiltration (penetration) regarding the horizontal plane.

The extrapolation of information is done by transferring parameters. Hydrological models can simulate ungauged basins by transferring parameters with higher accuracy using available hydrometeorological data (*Dastorani & Poormohammadi, 2012*). Additionally, basin characteristics, such as soil type and its particular traits, must be considered. The SMM divides the basins according to land use. Then parameters are associated with each type of soil, which can be modified monthly without cyclical variations at the annual level.

Basin nesting refers to a watershed that is hydrologically connected to another if the riverbed runoff discharges from the previously modeled basin. There will be some areas with information from two basins, which will provide data on the sites of their discharge.

The model results will be as reliable as the assumptions, the available input data, and the estimated parameters.

The model is calibrated by the modification of some parameters that represent the behavior of the basin, the calibration must be statistical and graphic. In the case of rainfall-runoff models (RRM), it is done by applying optimization techniques that seek a set of parameters that cause the model results to match the observed values as closely as possible.

Calibration with WEAP is performed manually, which consists of an iterative trial and error process. Each time the model parameters are adjusted, the model results and historical inputs are compared using goodness-of-fit indicators and graphs generated with the observed and simulated series.

Four goodness-of-fit indicators are used to calibrate the surface modeling: the Nash-Sutcliffe efficiency (NSE) (*Nash & Sutcliffe, 1970*) is a mean square error that gives greater weight to considerable mistakes, which often, but not always, occur during periods of high flow. The modified Nash-Sutcliffe efficiency (ln NSE) (*Muleta, 2012*) considers the logarithmic transformation of the flow and gives greater weight to errors during low flows. Moreover, Pearson's correlation coefficient (r) (*Muleta, 2012*) measures the covariance of observed and simulated values without bias penalty. And the Symmetry coefficient (SC) (*Muleta, 2012*) is a measure of the symmetry of the fit between the mean simulation and the mean observation; this characteristic, although simple, is critical to preserve in long-period modeling.

The quality of the indicators depends on the value obtained in the calibration, so according to *Moriasi et al. (2007)*, if the value is lower than 0.50, the quality of the modeling is unsatisfactory; the minimum value expected with the calibrations is 0.50 to 0.60 to have satisfactory modeling; however, the desired values are from 0.60 to 0.70 and above 0.70 for good and very good calibrations, respectively.

## RESULTS AND DISCUSSION

The results are presented according to the sections described in the methodology. These include the analysis and validation of meteorological and hydrometric stations, isolated modeling (for individual basins), and grouped modeling through the transfer of parameters in nested basins.

The meteorological stations are selected based on criteria such as the percentage of gaps, the number of years under study, as well as the splicing of hydrometric information.

A review of the stations located within a buffer of 30 km of the study area was carried out, resulting in a total of 77 stations. Several stations were discarded until 44 were found to be operating, to which homogeneity and independence tests were carried out. Finally, according to the results of the tests, a total of 30 meteorological stations (shown in Table 1) were used for the surface modeling.

Regarding the review of hydrometric stations, a radius of influence of 50 km was created for the study area periphery. A total of 75 hydrometric stations were obtained, and the data from these stations was downloaded and processed to access basic information, such as years in service, years with data, longest continuous series, and the percentage of gaps. Based on these criteria, 61% of the stations were eliminated, leaving only 29 stations.

From the remaining stations, the relevant graphs were created for trends or leaps. Consistency tests were also applied, in which only 12 stations were sampled.

For the final selection of the stations, the calculation of the runoff coefficient (RC) and the relative modulus (RM) were considered, which should only fulfill the application intervals.

Table 2 shows the results for the four basins that were modeled in an isolated manner. Homogeneity and persistence tests were applied to these stations to ensure the reliability of the data. In particular, the hydrometric stations are located within a 20 km buffer.

The basin was generated from hydrometric station 12415 Puente San Isidro. It is the most representative basin and has adequate information for analysis. As shown in Table 2, it complies with the parameters of RM and RC within the recommended range, as well as with the consistency tests.

Concerning the review of the annual graphs (Fig. 5), there is a downward trend, in addition to a downward leap starting in 1980. Therefore, there are two well-established periods (1969–1980 and 1980–1989), in which the difference between the averages of both periods is 5.17 hm$^3$/year.

The model assembly is shown in Fig. 6, where the elements required to obtain information on both the basin and the recharge that may be available in the area can be observed. The main parameters such as hydrometric station (HS), river (R), basin (B), and aquifer (GW) are displayed, as well as the basin-river (to represent runoff) and basin-aquifer (to represent infiltration) connections.

For basin modeling, the first phase is the calibration of evapotranspiration, which compares the mean values using the Thornthwaite method against the real (actual) values calculated by WEAP. In this case, the mean evapotranspiration obtained by the

**Table 1 Meteorological stations used for modeling.**

| Code | Name | Service years | Efective years | Empty data percent | Anual mean precipitation |
|---|---|---|---|---|---|
| 16002 | Agostitlán | 56 | 52.2 | 6.8% | 1,336.28 |
| 16022 | Cointzio | 66 | 60 | 9.1% | 811.44 |
| 16028 | Cuitzillo Grande | 38 | 36.5 | 3.9% | 654.85 |
| 16045 | El Temazcal | 49 | 48.3 | 1.4% | 1,309.04 |
| 16055 | Jesús del Monte | 79 | 76.6 | 3.0% | 867.08 |
| 16080 | Morelia (SMN) | 29 | 27.4 | 5.5% | 736.10 |
| 16081 | Morelia (DGE) | 68 | 67.1 | 1.3% | 781.68 |
| 16087 | Pátzcuaro | 46 | 39.8 | 13.5% | 954.46 |
| 16096 | Malpaís | 74 | 59.2 | 20.0% | 713.52 |
| 16105 | Quirio, Indaparapeo | 52 | 49.2 | 5.4% | 693.89 |
| 16109 | San Diego Curucpatle | 93 | 90.2 | 2.9% | 1,050.30 |
| 16136 | Tzitzio | 46 | 39.7 | 13.7% | 1,234.31 |
| 16235 | Huajumbaro | 35 | 31.2 | 10.9% | 1,171.34 |
| 16250 | Huandacareo | 32 | 26.6 | 16.9% | 908.90 |
| 16254 | Teremendo | 33 | 32.5 | 1.5% | 693.71 |
| 11002 | Acámbaro | 79 | 75.4 | 4.6% | 753.95 |
| 16052 | Huingo | 74 | 72.3 | 2.3% | 739.25 |
| 11077 | Tarandacuao | 74 | 71.5 | 3.4% | 767.92 |
| 16027 | Cuitzeo | 92 | 82.9 | 9.9% | 669.22 |
| 16145 | Zinapécuaro | 92 | 81.3 | 11.6% | 804.57 |
| 16050 | Huaniqueo | 66 | 64.8 | 18.2% | 853.54 |
| 11072 | Santa Rita | 55 | 54.7 | 5.5% | 706.24 |
| 11060 | Salvatierra | 79 | 71.7 | 9.2% | 719.59 |
| 11071 | Santa María (DGE) | 76 | 69.9 | 8.0% | 673.31 |
| 11076 | Presa Solís | 55 | 52.2 | 5.1% | 725.85 |
| 11010 | Cerano | 54 | 50.5 | 6.5% | 719.83 |
| 16016 | Carrillo Puerto | 45 | 42 | 6.7% | 696.58 |
| 16155 | Copándaro | 40.9 | 34.6 | 15.5% | 816.99 |
| 16084 | Panindícuaro | 68.9 | 39.1 | 43.3% | 799.34 |
| 16142 | Zacapu | 42.2 | 33.5 | 20.6% | 872.63 |

**Table 2 Information from the hydrometric stations with the calibration period.**

| Code | Name | A (km²) | R (hm³/year) | P (mm/year) | RC | RM | Dams | PC |
|---|---|---|---|---|---|---|---|---|
| 12314 | Queréndaro | 134.26 | 37.55 | 1,200 | 0.23 | 8.9 | 0 | 1975–1986 |
| 12347 | Santiago Undameo | 623.12 | 68.98 | 900 | 0.12 | 3.5 | 2 | 1951–1987 |
| 12415 | Puente San Isidro | 216.17 | 18.36 | 900 | 0.09 | 2.7 | 2 | 1969–1988 |
| 12620 | Tarímbaro | 94.52 | 10.36 | 900 | 0.12 | 3.5 | 0 | 1979–1986 |

**Note:**
A, area; R, runoff; P, precipitation; RC, runoff coefficient; RM, relative modulus; Dams, number of dams in the basin; PC, period of calibration.

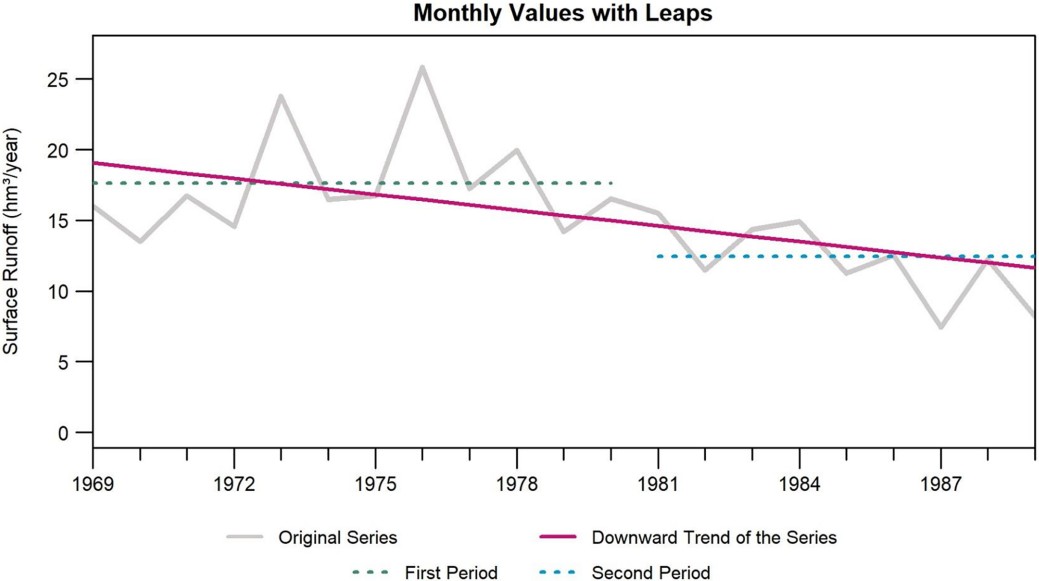

**Figure 5 Monthly volumes of hydrometric station 12415 with leaps.** Created with RStudio (R version 4.2.1; *RStudio Team, 2022*; *R Core Team, 2022*).     

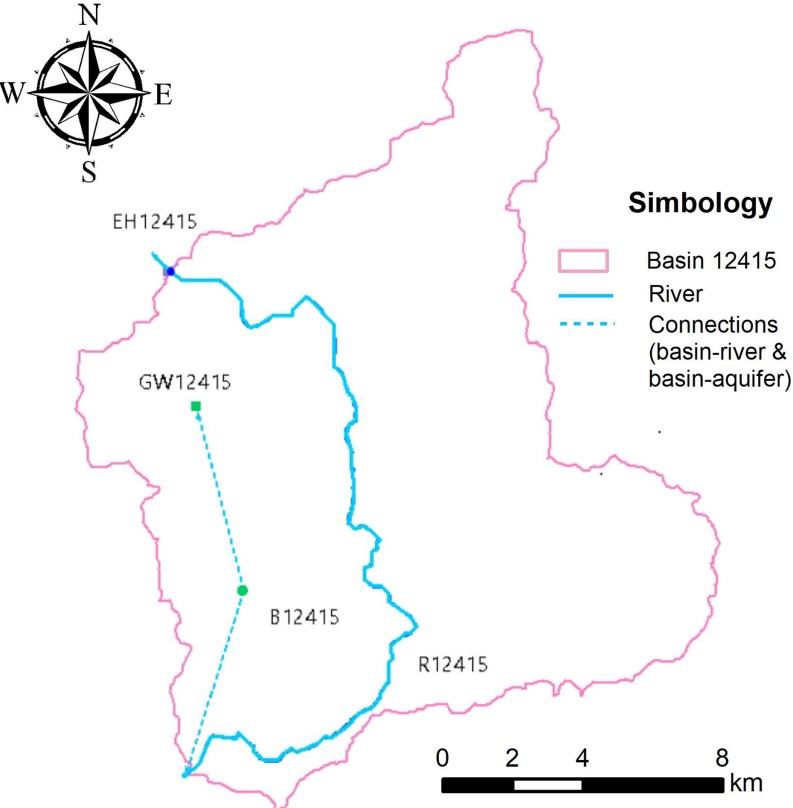

**Figure 6 Modeling scheme of basin 12415 Puente San Isidro in WEAP.** R, river; B, basin; GW, aquifer; HS, hydrometric station. Created with QGis3.18.     

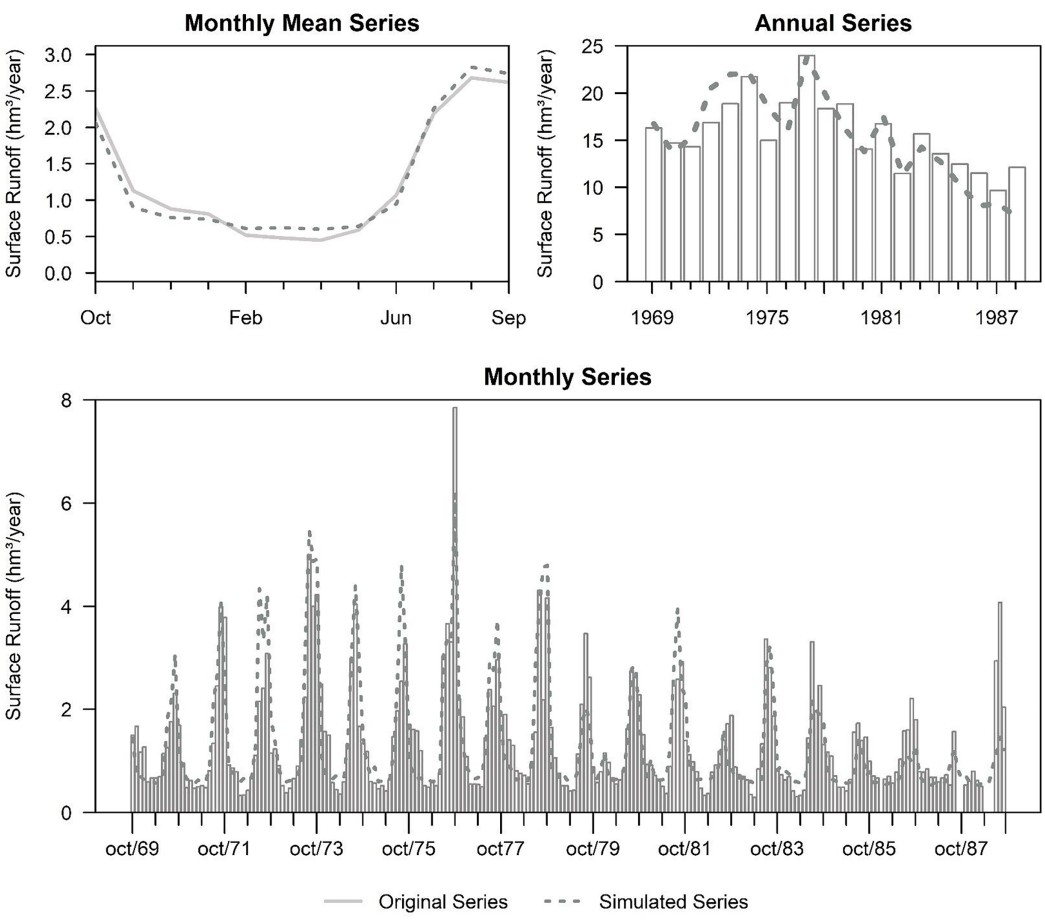

**Figure 7 Visual calibration from the modeling of basin 12415.** Created with RStudio (R version 4.2.1; *RStudio Team, 2022*; *R Core Team, 2022*).

**Table 3 Results of the surface modeling goodness-of-fit indicators.**

| Goodness-of-fit indicators | Basin | | | |
| --- | --- | --- | --- | --- |
| | 12314 | 12347 | 12415 | 12620 |
| Nash Sutcliffe | 0.50 | 0.66 | 0.65 | 0.55 |
| Ln Nash Sutcliffe | 0.62 | 0.81 | 0.73 | 0.51 |
| Pearson | 0.71 | 0.82 | 0.82 | 0.75 |
| Symmetry coefficient | 0.99 | 0.99 | 0.99 | 0.99 |

Thornthwaite method is 65.04 mm, and that estimated with WEAP is 59.78 mm, providing an error of 5.26 mm (8.08%).

The values are correctly adjusted for the graphical calibration of basin 12415 Puente San Isidro. Although there is no exact trend, the simulated volumes are compensated below and above the observed ones Fig. 7-Monthly Mean Series, achieving satisfactory goodness-of-fit indicators (Table 3). In Fig. 7-Annual Series, on the other hand, the adjustment in the first years of the modeling is fulfilled in trend but not precision since, until 1974, there was
**Table 4 Effective parameters (Kc, SWC, RRF) used for basin 12415.**

|  | Land use | | | | | Single value |
|---|---|---|---|---|---|---|
| **Kc** | Agricultural | | | | | 0.80 |
|  | Urban | | | | | 0.60 |
|  | Forest | | | | | 0.80 |
|  | Grassland/Scrubland | | | | | 0.70 |

**Monthly values**

| **SWC** | **Land use** | **October** | **November** | **December** | **January** | **February** | **March** |
|---|---|---|---|---|---|---|---|
|  | Agricultural | 1,700 | 1,700 | 1,700 | 1,700 | 1,700 | 1,600 |
|  | Urban | 1,300 | 1,300 | 1,300 | 1,300 | 1,300 | 1,200 |
|  | Forest | 1,550 | 1,550 | 1,550 | 1,550 | 1,550 | 1,450 |
|  | Grassland/Scrubland | 1,400 | 1,400 | 1,400 | 1,400 | 1,400 | 1,300 |
|  | **Land use** | **April** | **May** | **June** | **July** | **August** | **September** |
|  | Agricultural | 1,600 | 1,600 | 1,600 | 1,600 | 1,700 | 1,700 |
|  | Urban | 1,200 | 1,200 | 1,200 | 1,200 | 1,300 | 1,300 |
|  | Forest | 1,450 | 1,450 | 1,450 | 1,450 | 1,550 | 1,550 |
|  | Grassland/Scrubland | 1,300 | 1,300 | 1,300 | 1,300 | 1,400 | 1,400 |
| **RRF** | **Land use** | **October** | **November** | **December** | **January** | **February** | **March** |
|  | Agricultural | 4 | 3.7 | 4.3 | 4.5 | 4.5 | 4.45 |
|  | Urban | 2 | 1.7 | 2.3 | 2.5 | 2.5 | 2.45 |
|  | Forest | 4 | 3.7 | 4.3 | 4.5 | 4.5 | 4.45 |
|  | Grassland/Scrubland | 3 | 2.7 | 3.3 | 3.5 | 3.5 | 3.45 |
|  | **Land use** | **April** | **May** | **June** | **July** | **August** | **September** |
|  | Agricultural | 4.45 | 4.5 | 4.2 | 3.8 | 4 | 4.3 |
|  | Urban | 2.45 | 2.5 | 2.2 | 1.8 | 2 | 2.3 |
|  | Forest | 4.45 | 4.5 | 4.2 | 3.8 | 4 | 4.3 |
|  | Grassland/Scrubland | 3.45 | 3.5 | 3.2 | 2.8 | 3 | 3.3 |

an adjustment due to the warm-up period. Finally, as seen in Fig. 7-Monthly Series, the simulated values are above the observed values, but the monthly volumes are accurately represented. For this basin, a base discharge of 0.5184 hm$^3$ per month is also considered.

The procedure shown for basin 12415 was performed for the remaining four basins (Fig. 1), which are graphically and statistically calibrated according to the calibration periods shown in Table 2.

Table 3 shows the summary of the statistical calibration of the four basins modeled in WEAP, in which it is observed that the parameters are considered as good.

The effective parameters to be calibrated (Kc, SWC, f, and RRF) are modified for each of the modeled basins shown Table 4. These depend on the use and type of soil. Monthly relationships can be established so the values will have monthly variations.

 

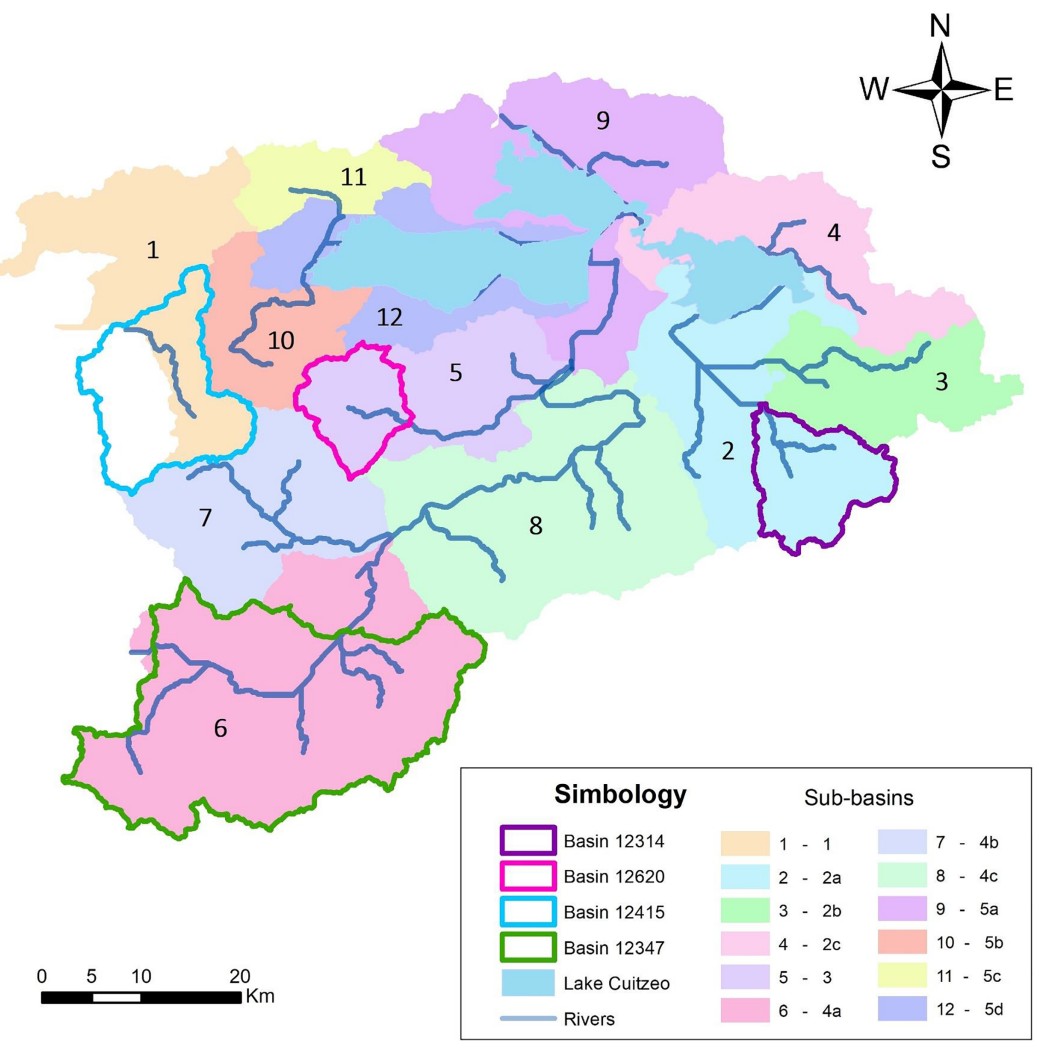

**Figure 8 Division of the 12 modeling sub-basins and location of the four isolated basins modeled in WEAP.** Created with QGis3.18.

The grouped modeling of the study area was divided into 12 sub-basins (Fig. 8) and categorized into five zones, in which a transfer of parameters was carried out. Through this process, it was relatively simple to model the entire area. In addition, the decision to work with 12 sub-basins was necessary to have greater detail in the distribution of meteorological information. The discretization of the sub-basins into 2-a, 2-b and 2-c is due to the fact that these are nested within the same hydrological network, which in turn generates sub-basin 2. The same happens with 4-a, 4-b and 4-c; as well as with 5-a, 5-b, 5-c and 5-d.

The modeling in WEAP is done grouped, and the generated scheme is shown in Fig. 9, where the base elements (hydrometric station, river-basin, and basin-aquifer relationship) are indicated for each of the 12 sub-basins. The grouped modeling is supported by 21 meteorological stations (presented according to the influence in each basin).

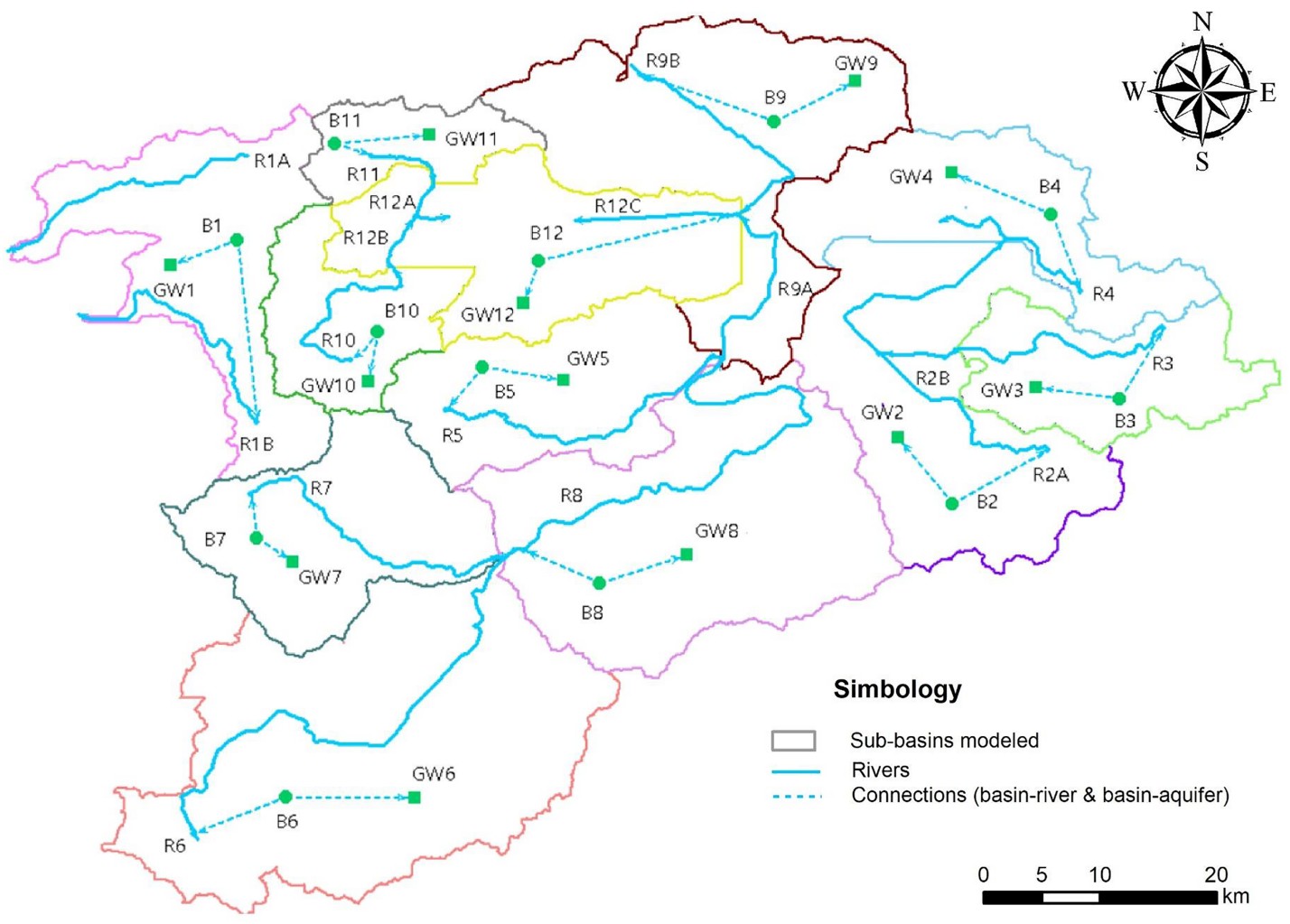

**Figure 9 Methodological scheme of the clustered modeling in WEAP.** R, river; B, basin; GW, aquifer; HS, hydrometric station. Created with QGis3.18.

Any existing relationship must be assessed between the basins for the transfer of parameters, whether they are nested in a larger watershed or part of the zone.

It is considered that the watersheds of zone 1 are nested in basin 12415 Puente San Isidro (which is the corresponding headwater basin).

Basin 12314 Queréndaro is at the headwaters of zone 2. Zone 3 is nested with basin 12620 Tarímbaro, which shares the same main riverbed. For zone 4, located to the south of the aquifer, the parameters used are those of basin 12347 Santiago Undameo, placed at the head of the zone, and the parameters are transferred to the neighboring watersheds that discharge into the same riverbed.

Finally, zone 5 cannot be nested to any modeled basin, but it is near basins 12620 Tarímbaro and 12347 Santiago Undameo. The latter connects the river source to Lake Cuitzeo (Fig. 8).

As shown in Fig. 10, with the acceptable parameters, modeling can be performed according to the availability of meteorological information to extend the modeled series to
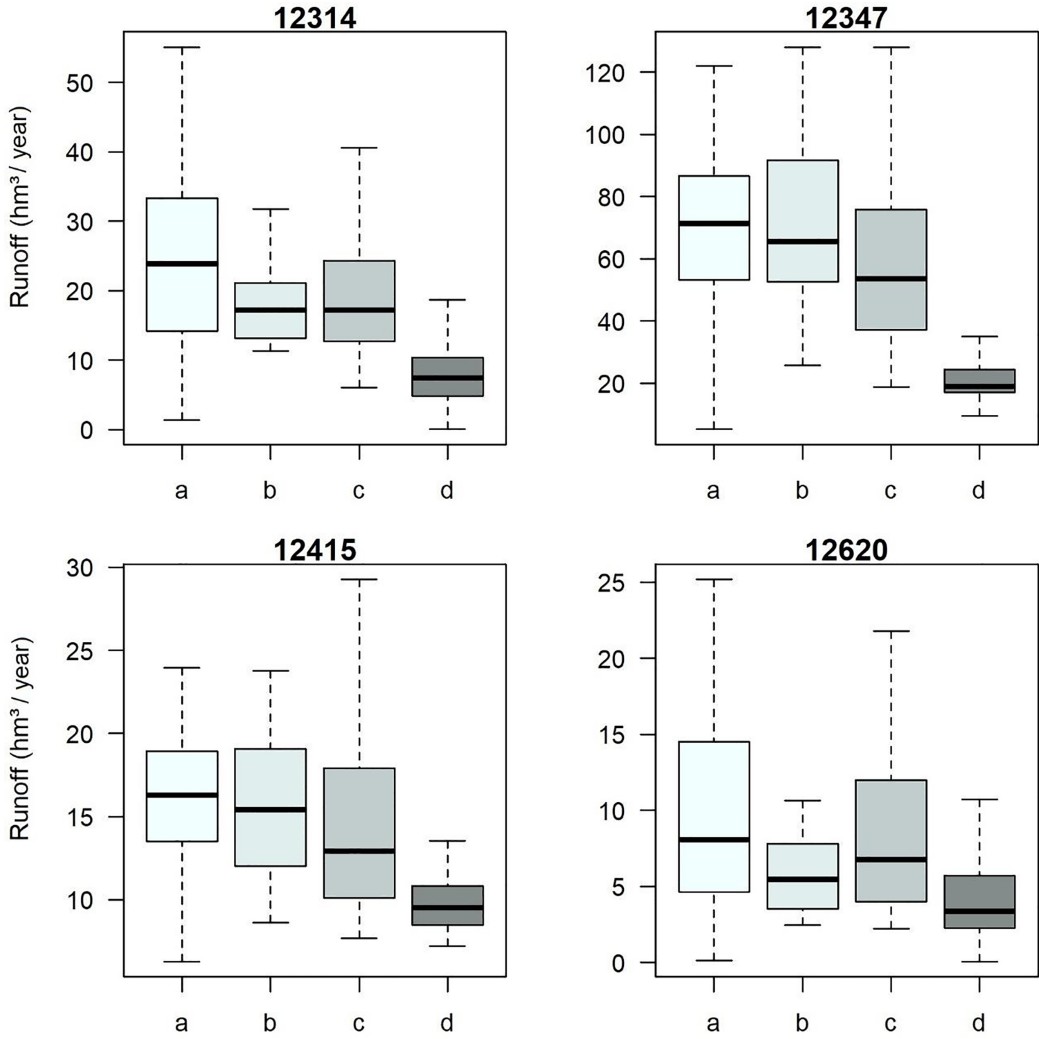

**Figure 10 Sub-basin modeling being (A) original series, (B) calibration period, (C) extended period, and (D) without treatment.** Created with RStudio (R version 4.2.1; *RStudio Team, 2022*; *R Core Team, 2022*).

any period of interest. In the same way, it can be applied to different scenarios and any generated basin that is nested to it. Moreover, surface modeling using untreated meteorological stations results in runoff volumes lower than those observed, thus modifying their statistical parameters, these are limited to representing 25% to 50% of the runoff recorded at the hydrometric stations.

## CONCLUSIONS

Management of meteorological information is essential. These series are the input to the surface model, so if they are not carefully reviewed, we could be entering poor-quality information that will provide a result but may not be factual or representative of the basin.

The graphic review of the hydrometric stations is also fundamental because modeling must be carried out for basins in a natural regime. If a basin is in an altered regime,

restitution to a natural regime must be reached, which complicates the process due to the scarcity of data.

It is important to know the behavior of the riverbed present in the area, which allows us to understand how they nest according to the hydraulic functioning of the basin.

Model calibration is crucial in areas with a temporal shortage of information because the parameters shift spatially and temporally (to more current periods), so the errors we can accumulate are meaningful.

As seen throughout the process, different methods are compiled, which helps the series to be continuously modified and reviewed throughout the methodology, ensuring the correct selection of information.

This methodology was correctly applied in the four modeled basins, thus providing a general validation, which is relevant for areas of LATAM that have a scarcity of temporal and spatial information.

The modeling presented temporally and spatially extrapolated information.

The modeled basins had calibration periods ranging from 7 to 36 years. However, with the transfer of parameters and the nesting of watersheds, it was possible to extend the modeling to 76 years, with an average growth in temporal information of 400%.

A total surface area of 1,068 km$^2$ of isolated basins was available concerning the modeled surface area. Likewise, it was possible to have information from basins covering 4,290 km$^2$ of surface area, which represents an increase in spatial data of approximately 400%.

For the temporally extended modeling, and according to the established methodology, the mean was preserved, as well as extreme values close to the observed runoff values.

This methodology is limited to having the minimum information necessary to perform statistical tests for the analysis and validation of meteorological and hydrometric series.

## ACKNOWLEDGEMENTS

We value the moral support provided by the Universidad Michoacana de San Nicolás de Hidalgo, especially the Department of Chemical Engineering and the Hydraulics Department.

### Funding

The authors received no funding for this work. María del Mar Navarro-Farfán received a scholarship for her Ph.D. studies from the National Council for Humanities, Science and Technology of Mexico (Conahcyt). The funders had no role in study design, data collection and analysis, decision to publish, or preparation of the manuscript.

### Grant Disclosures

The following grant information was disclosed by the authors:
National Council for Humanities, Science and Technology of Mexico (Conahcyt).

## Competing Interests

The authors declare that they have no competing interests.

## Author Contributions

- María del Mar Navarro-Farfán conceived and designed the experiments, performed the experiments, analyzed the data, prepared figures and/or tables, authored or reviewed drafts of the article, and approved the final draft.
- Liliana García-Romero conceived and designed the experiments, performed the experiments, analyzed the data, prepared figures and/or tables, authored or reviewed drafts of the article, and approved the final draft.
- Marco A. Martínez-Cinco conceived and designed the experiments, authored or reviewed drafts of the article, and approved the final draft.
- Constantino Domínguez-Sánchez conceived and designed the experiments, authored or reviewed drafts of the article, and approved the final draft.
- Sonia Tatiana Sánchez-Quispe conceived and designed the experiments, authored or reviewed drafts of the article, and approved the final draft.

## Data Availability

The hydrometric and meteorological stations are available in the Supplemental Files.

## Supplemental Information

Supplemental information for this article can be found online at http://dx.doi.org/10.7717/peerj.17755#supplemental-information.

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
