# Peer review of "Methodology for the assessment of poor-data water resources"

_PeerJ, doi:10.7717/peerj.17755_

## Round 0.1 · original submission · Minor Revisions

Dear Authors,

I have reviewed the comments from the two reviewers for your manuscript titled "Methodology for the Assessment of Poor-Data Water Resources." Based on their feedback, my decision is to request a minor revision.

Please revise your manuscript according to the reviewers' suggestions and submit it for further review.

Regards,
Seyed Mohamamd Moein Sadeghi

Reviewer 1 ·

Basic reporting

English is clear.
there is no consistency in citing references. for instance sometimes "&" and sometimes "and" is used in citation.

Experimental design

it is ok.

Validity of the findings

it is acceptable.

Additional comments

Formatting and Consistency:

Please use the justify option in Word for the entire text.
Ensure consistent use of "and" vs. "&" for citing authors.

Abstract:
It is better to have the abstract as one paragraph instead of separating it into three separate paragraphs for Background, Methods, and Results.

Line 103: Please indicate the calibration points in Figure 1 and explain what "HS" stands for in the figure caption.
Line 236: The wording "concerning the information to be obtained from the model" is vague. Modify it to "a sensitivity analysis must be completed to determine the influence of each parameter on the model output."
Line 263: "Nash-Sutcliffe efficiency" – remove the space before and after the hyphen throughout the text.
Line 324: Clarify the time units used for discharge. Ensure all discharge units are correct throughout the text.
Figures and Tables:

Figure 1: In the caption, explain what "HS" stands for (HS stands for Hydrometric Station).
Figures 5, 7, and 10: Correct the unit of runoff. Specify the period (e.g., (hm^3)/yr). Ensure the unit is correct in Table 1 as well.
Figures 6 and 9: Add north direction, scale, and legend.
Figure 8: Clarify what sub-basins 2a, 2b, 4a, etc., represent.

Reviewer 2 ·

Basic reporting

The introduction effectively outlines the context and significance of the study, providing a comprehensive overview of previous research and the knowledge gap addressed. Also, the literature review is thorough and relevant. Specific suggestions include emphasizing the novelty of the methodology in the introduction, summarizing complex procedures in the methodology section, and addressing minor language issues.

Experimental design

The research question is clearly defined, relevant, and addresses a significant knowledge gap in hydrologic modeling with limited data. The methods are described with sufficient detail, allowing for replication. However, the manuscript could benefit from a more explicit explanation of the decision-making process in selecting meteorological and hydrometric stations. Overall, the experimental design is robust, well-detailed, and contributes meaningfully to the field.

Validity of the findings

The conclusions are clearly stated, directly linked to the original research question, and supported by the results. However, the manuscript would benefit from a more explicit discussion on the limitations of the study, particularly regarding the generalizability of the findings to other regions with different hydrological characteristics. Additionally, while the study provides a strong case for the proposed methodology, further validation through independent datasets or case studies could enhance the credibility and impact of the findings.

Additional comments

The study focuses on a specific region (Morelia - Queréndaro aquifer in Mexico), which may limit the generalizability of the findings. A discussion on how the methodology can be adapted to other regions with different hydrological and climatic conditions would be beneficial. The manuscript would benefit from a more explicit discussion of the study’s limitations. For instance, the potential biases introduced by data gaps and the reliance on certain statistical methods should be addressed. Minor language issues need attention to ensure clarity and readability. Professional editing could help address awkward phrasing and complex sentence structures. in the end, using multiple statistical tests and goodness-of-fit indicators adds robustness to the findings, and the detailed analysis of meteorological and hydrometric data is commendable.

---

## Round 0.2 · accepted · Accept

Dear Authors,

I am pleased to inform you that after careful consideration of the revised version of your manuscript and your responses to the reviewers' comments, I have decided to accept your manuscript for publication. I appreciate the effort you put into addressing all the concerns raised during the review process. Your thorough and thoughtful revisions have significantly enhanced the manuscript, making a valuable contribution to the field.

Congratulations on your hard work and thank you for your diligence in improving the manuscript. We look forward to publishing your research and sharing it with the wider community.

Best wishes,
Seyed Mohammad Moein Sadeghi